# Genomic Characterization and Molecular Evolution of Sapovirus in Children under 5 Years of Age

**DOI:** 10.3390/v16010146

**Published:** 2024-01-19

**Authors:** Xiaolei Ji, Chen Guo, Yaoyao Dai, Lu Chen, Yujia Chen, Shifang Wang, Yihua Sun

**Affiliations:** 1Key Laboratory of Medicine, Nantong Center for Disease Control and Prevention, 189 Gongnongnan Road, Chongchuan District, Nantong 226007, China; ntcdcjxl@outlook.com (X.J.); daiyaoyao2010@126.com (Y.D.); 15190871517@163.com (L.C.); chenyujia0312@163.com (Y.C.); 2Department of Laboratory Medicine, Nantong Chongchuan Center for Disease Control and Prevention, 47 Zhongxiu Middle Road, Nantong 226001, China; gc901102@sina.com

**Keywords:** sapovirus, GI.1, GI.6, molecular evolutionary rate, amino acid variations, population dynamics

## Abstract

Sapovirus (SaV) is a type of gastroenteric virus that can cause acute gastroenteritis. It is highly contagious, particularly among children under the age of 5. In this study, a total of 712 stool samples from children under the age of 5 with acute gastroenteritis were collected. Out of these samples, 28 tested positive for SaV, resulting in a detection rate of 3.93% (28/712). Samples with Ct < 30 were collected for library construction and high-throughput sequencing, resulting in the acquisition of nine complete genomes. According to Blast, eight of them were identified as GI.1, while the remaining one was GI.6. The GI.6 strain sequence reported in our study represents the first submission of the GI.6 strain complete genome sequence from mainland China to the Genbank database, thus filling the data gap in our country. Sequence identity analysis revealed significant nucleotide variations between the two genotypes of SaV and their corresponding prototype strains. Phylogenetic and genetic evolution analyses showed no evidence of recombination events in the obtained sequences. Population dynamics analysis demonstrated potential competitive inhibition between two lineages of GI.1. Our study provides insights into the molecular epidemiological and genetic evolution characteristics of SaV prevalent in the Nantong region of China, laying the foundation for disease prevention and control, as well as pathogen tracing related to SaV in this area.

## 1. Introduction

Sapovirus (SaV) is a type of gastroenteric virus that can cause acute gastroenteritis (AGE). The infection of SaV is also considered a significant global public health issue [1]. Similar to Norovirus (NoV), SaV primarily spreads among populations through contaminated water and food. SaV mostly causes sporadic infections, but it can also cause outbreaks in relatively closed environments such as schools, hospitals, childcare centers, and nursing homes [2,3]. The population is generally susceptible to SaV, with the highest prevalence in children under 5 years old [4]. SaV infections usually present with mild symptoms, but some rare cases may require hospitalization [5]. To date, an effective vaccine for prevention and control has not been developed.

SaV belongs to a genus in the caliciviridae family. Its genome is a non-segmented, positive-sense RNA molecule of approximately 7.5 kb in length, with a poly(A) tail at the 3’-end. The human-infecting SaV genome contains two open reading frames (ORFs), namely ORF1 and ORF2. ORF1 encodes a polyprotein and the major capsid protein VP1, while ORF2 encodes the minor capsid protein VP2 [6]. The polyprotein can be cleaved by a protease into non-structural proteins NS1~NS7. Among these, the functions of NS3, NS5, NS6, and NS7 have been confirmed. NS3 functions as a helicase, NS5 serves as a primer for viral genome replication, NS6 is involved in proteolysis, and NS7 acts as a polymerase in the viral genome [7,8,9]. However, the functions of NS1, NS2, and NS4 currently remain uncertain. Research suggests that these proteins may be associated with tissue tropism, virulence, and epidemiological fitness [10]. Based on the complete VP1 sequence, SaV can be classified into 19 genogroups, designated as GI to GXIX [11]. Among them, GI, GII, GIV, and GV are known to infect humans. These four genogroups can further be divided into 18 genotypes [12,13], including GI.1 to GI.7, GII.1 to GII.8, GIV.1, and GV.1 to GV.2. In the research report by Oka T et al., it was found that the cutoff values for genotype and genogroup clusters were designated as ≤0.169 and ≤0.488, respectively, based on nucleotide sequences. Similarly, based on amino acid sequences, two major peaks (0 to 0.480 and 0.652 to 1.115) were observed, corresponding to the strain and genogroup range, respectively [14].

Although SaV has been recognized as one of the causes of AGE, there have been very few reports on the genetic characteristics and evolutionary analysis of SaV whole genomes based in China. This study reports nine SaV whole genome sequences obtained from fecal samples of children under 5 years old in the Nantong area of China. We compared these sequences with those reported by other researchers and conducted evolutionary analysis. The aim of this study is to provide scientific basis for the prevention and control of SaV viruses, and also to characterize and provide the sequence information of the first full-length genome of SaV GI.6 in mainland China, thereby providing a foundation for future research.

## 2. Materials and Methods

### 2.1. Source of the Specimen

The specimens used in this study were obtained from the designated monitoring hospital in Nantong, China, for children under 5 years old with AGE and diarrhea in the emergency department. A total of 712 fecal specimens were collected from July 2020 to June 2022. After collection, the specimens were promptly transported to the laboratory at low temperature to complete the pathogen monitoring work and subsequent experiments.

### 2.2. RNA Extraction and SaV RT-qPCR Detection

A 10% (*m*/*v*) fecal suspension was prepared by mixing an appropriate amount of a phosphate buffer (pH 7.2~7.4) with feces. The mixture was vortexed for 5 min and then centrifuged at 12,000× *g* at 4 °C for 15 min to collect the supernatant. Nucleic acids were extracted from 300 μL of the supernatant using a fully automatic nucleic acid extractor (Liferiver, Shanghai, China, Code No. EX3600Plus) and nucleic acid extraction reagent (Liferiver, Code No. Z-M-0092-96) following the provided instructions. For the RT-qPCR detection of the extracted nucleic acid, the reagent of choice (Takara, Beijing, China, Code No. RR064A) was used, as suggested by Maarseveen et al. [15], along with SaV universal primers and probes (SAPOs, SAPOas, SAPO-XS-QS705). The thermal cycling conditions were as follows: 42 °C for 5 min; 95 °C for 10 s; 95 °C for 5 s; and 60 °C for 34 s, repeated for 40 cycles. Fluorescence signals were collected at 60 °C, and a Ct value ≤ 38 with a clear sigmoid amplification curve was considered positive. SaV nucleic acid samples with positive results and a Ct value < 30 were transferred to sterile, enzyme-free EP (Eppendorf) tubes and stored at −80 °C in the freezer.

### 2.3. High-Throughput Sequencing, Generation of Virus Genome Sequences

We chose the VAHTS Universal V8 RNA-seq Library Prep Kit for Illumina (Vazyme, Nanjing, China, Code No. NR605) and strictly followed the instructions to construct RNA libraries from nucleic acid samples with Ct values < 30. After library construction, we performed initial quantification using Qubit 4.0 and diluted the libraries to a concentration of 1 ng/μL. Next, we used Agilent 2100 to detect the length of library insert fragments. Then, we utilized the VAHTS Library Quantification Kit for Illumina (Vazyme, Code No. NQ103) for accurate quantification of the library’s effective concentration and diluted the libraries to a concentration of 4 nM. Sequencing carried out using the Miseq system (Illumina, USA) paired with the Miseq Reagent kit v3 600 Cycles PE (Illumina, Code No. 15043895). The sequencing strategy would be paired-end sequencing with a read length of 301 bp for each end. The full-length genome sequence of SaV was generated using the self-built workflow in CLC Genomics Workbench V22.0.1. The workflow involved several steps. Firstly, the Trim Reads program was used to remove bases with low quality. The Quality limit was set at 0.05 and the Maximum number of ambiguities was set at 2. After that, the Map reads to Reference program was utilized to align the data with reference genomes of each SaV genotype. This alignment process ultimately yielded the full-length genome sequence of SaV.

### 2.4. Genotyping of the SaV

We submitted the obtained full genome sequence of SaV to the web-based Human Calicivirus Typing tool (https://calicivirustypingtool.cdc.gov/) (accessed on 20 August 2023) [16] for virus genotype identification. And we used Geneious Prime v2023.0.4 (https://www.geneious.com/) (accessed on 20 August 2023) to annotate the obtained full-length genome sequence of SaV with reference sequences and uploaded it to the Genbank database.

### 2.5. Analysis of Sequence Identity, Phylogeny, and Genetic Evolution

We used BioAider v1.521 [17] to perform consistency analysis on these nine sequences with the original strains of SaV GI.1 (HM002617) [18] and GI.6 (AJ606694) [19], and create a similarity matrix. Then, we selected representative strains of different subtypes of SaV from Genbank and used the maximum likelihood method (ML) to construct the systematic phylogenetic tree for the complete genome, non-structural protein (NSP) region, VP1 gene, and VP2 gene of the virus. The GTR + G + I model was chosen for model selection, with a bootstrap value set at 1000. We checked and downloaded all whole genome reference sequences for GI.1 and GI.6 from Genbank. And we performed multiple sequence alignment using MAFFT [20]. For sequence recombination analysis, we utilized RDP4 [21] software.v4.101. The sequences used in the analysis were all complete genome sequences of SaV obtained from Genbank. The molecular clock genetic evolutionary analysis was conducted using the BEASTv1.10.4 [22], based on the MCMC (Markov chain Monte Carlo) algorithm, to calculate the molecular evolutionary rate and tMRCA (time to the Most Recent Common Ancestor) of the whole genome sequence mentioned above. We used the ModelFinder [23] function in IQ-Tree [24] to estimate nucleotide substitution models for each branch, and selected the model with the highest AIC score (see Table 1). All types of molecular clocks were selected to be uncorrelated relaxed clocks [25], with coalescent:Bayesian skyline, and set at 1,000,000,000 sampling for every 100,000 states; the effectiveness of the posterior distribution for each setting was evaluated based on the effective sample size (ESS) using Tracer v 1.7 [26], with 10% burning. A minimum ESS value of 200 was considered acceptable. We constructed a Bayesian Skyline Plot (BSP) to infer the dynamic changes in the effective population size of SaV over time. To generate the MCC (Maximum Clade Credibility Tree), we used TreeAnnotator v1.10.4. Then, the output was beautified using the web application tvBOT [27] for phylogenetic trees. After that, in order to identify the key amino acid substitution sites which caused the differentiation of GI.1 SaV into two lineages, we used MEGA 11 [28] to translate the various structural and non-structural proteins of GI.1 SaV mentioned above.

## 3. Results

### 3.1. Prevalence of SaV

In this study, a total of 28 out of 712 stool specimens tested positive for SaV, resulting in a positivity rate of 3.93% (28/712). As shown in Figure 1, The peak positivity rates in both monitoring years were observed in November, with rates of 16.67% and 12.50%, respectively. Throughout the two monitoring years, positive specimens were detected in both the first and fourth quarters, with a higher positivity rate in the fourth quarter compared to the first quarter. However, no positive specimens were detected in the second and third quarters of each year. These findings suggest a clear seasonal distribution pattern for SaV infections.

### 3.2. SaV Sequencing Results and Genotype Identification

For RNA library construction, this study selected a total of 15 samples with Ct values < 30. The Illumina NovaSeq sequencing platform was used for sequencing, resulting in an average output data of 3.44 Gb per sample. All samples had a Q30 value greater than 95%. After mapping the sequencing data with reference sequences of different genotypes, a total of nine full-length SaV genomes were obtained. The average sequencing depth of the nine SaV genomes obtained in this study was higher than 100×, indicating satisfactory sequencing quality. These nine sequences were then analyzed using the web-based tool for sapovirus genotyping, resulting in eight sequences being identified as GI.1 and one sequence being identified as GI.6. Gene annotation was performed on these nine sequences using Geneious Prime software v2023.0.4 with reference genomes of the respective types, and the results showed that the coding sequences in all nine sequences could be translated correctly. The nine sequences have been submitted to the Genbank database with the accession numbers: OR672551~OR672559.

### 3.3. Analysis of Sequence Identity and Amino Acid Variation Characteristics

The nucleotide identity among nine local strains is between 73.04% and 98.24%. When comparing the sequences of eight local GI.1 strains to the original GI.1 strain (HM002617), the nucleotide identity is between 90.09% and 90.48%. When comparing the sequences of local GI.6 strains to the original GI.6 strain (AJ606694), the nucleotide identity is 88.33%, as shown in Figure 2A. The identity analysis between local strains and corresponding prototype strains based on different protein coding regions shows that point mutations are the main method of gene mutations in SaV strains. No amino acid insertions were found, only a site amino acid deletion (the eight local GI.1 strains lack alanine at position 590 compared to HM002617) was found in the NS6-NS7 coding region of GI.1 strains. By comparing the amino acid variations in each protein coding region, we found that the non-structural protein NS3 is the most conserved, while NS1 is more prone to variation. Among the structural proteins, VP1 is more conserved than VP2.

### 3.4. Phylogenetic Analysis

To construct a phylogenetic tree using the maximum likelihood (ML) method, we analyzed the whole genome, NSP gene, VP1 gene, and VP2 gene of the strains. The analysis was performed using the GTR + G + I model with a bootstrap value of 1000, as shown in Figure 3. Our study found that the nine SaV sequences obtained clustered together on the phylogenetic tree based on the complete genome, NSP gene, VP1 gene and VP2 gene, indicating no evidence of recombination. Additionally, we used RDP v4.0.1 software to conduct recombination analysis on all complete SaV sequences in Genbank, which confirmed that the nine SaV sequences obtained in this study did not undergo recombination.

### 3.5. Genetic Evolutionary Analysis

To reconstruct the timeline of the SaV evolutionary process, this study used the MCMC algorithm to infer the molecular evolutionary rate of the SaV whole genome, and constructed MCC. As shown in Figure 4, in the MCC based on the SaV whole genome, the two genotypes discussed in this study are located on two different branches and have a relatively large genetic distance. We found that the GI.1 strain branches into two lineages, with the GI.1 prototype strain (HM002617) belonging to Lineage I, while the eight GI.1 strains obtained in this study belong to Lineage II. For the GI.6 SaV genotype, a clear lineage differentiation has not been observed yet due to the limited data available.

In this study, a total of 86 SaV strains were included, and their whole genome sequences were used to estimate the molecular evolutionary rate and the time to the most recent common ancestor (tMRCA). The average molecular evolutionary rate for the 86 virus strains was 2.099 × 10^−3^ substitutions/site/year (95% highest posterior density interval: 1.143 × 10^−3^~2.761 × 10^−3^), with a tMRCA estimated to be around 1604. For the 73 GI.1 SaV strains, the average molecular evolutionary rate was 2.092 × 10^−3^ substitutions/site/year (95% HPD: 1.484 × 10^−3^~2.739 × 10^−3^), and the tMRCA was estimated to be around 1949. The average molecular evolutionary rate for the 5 GI.6 SaV strains was 6.881 × 10^−3^ substitutions/site/year (95% HPD: 1.874 × 10^−5^~2.930 × 10^−2^), with a tMRCA estimated to be around 1953. For the 36 GI.1 Lineage I strains, the average molecular evolutionary rate was 1.603 × 10^−3^ substitutions/site/year (95% HPD: 6.597 × 10^−4^~2.737 × 10^−3^), and the tMRCA was estimated to be around 1959. The average molecular evolutionary rate for the 45 GI.1 Lineage II strains was 2.082 × 10^−3^ substitutions/site/year (95% HPD: 1.569 × 10^−3^~2.653 × 10^−3^), with a tMRCA estimated to be around 1974, as shown in Table 2.

### 3.6. Analysis of Amino Acid Variations between Two Lineages of GI.1 SaV

After analyzing the amino acid variations in the above 81 GI.1 strains of SaV according to different genes, we discovered that the major reason for the division of GI.1 SaV into two lineages was due to variations at 21 amino acid positions, as shown in Figure 5. These positions were located at amino acid 61 in the NS1 gene, amino acids 84 and 112 in the NS2 gene, amino acids 28, 89, and 169 in the NS3 gene, amino acids 14, 53, and 71 in the NS4 gene, amino acid 99 in the NS5 gene, amino acids 1, 61, 331, 495, and 611 in the NS6-NS7 gene, amino acids 99, 332, and 523 in the VP1 gene, and amino acids 144, 146, and 163 in the VP2 gene.

### 3.7. Analysis of Population Dynamics

We performed population dynamics analysis on 86 full genome sequences of SaV included in this study by constructing a Bayesian Skyline Plot (BSP). The results indicate that GI.1 SaV was relatively stable before 2010, then showed a decreasing trend in effective population size from 2010 to 2015, and then leveled off (Figure 6A). In Lineage I of GI.1 SaV, the effective population size was stable before 2005, and then showed an expanding trend from around 2003 to 2013 (Figure 6B). After that, it slightly decreased and leveled off. In Lineage II of GI.1 SaV, the effective population size showed a slow expanding trend before 2002, then a significant decrease from 2002 to 2013, followed by a small peak and a rapid leveling off (Figure 6D). The effective population size of GI.6 SaV remained relatively stable during the outbreak period (Figure 6C). Both types of SaV showed a stable trend in effective population size before 2000, and then a clear decreasing trend.

## 4. Discussion

Apart from norovirus and rotavirus, SaV is also another significant pathogen responsible for causing acute gastroenteritis in children in various countries and regions. In this study, we conducted a two-year monitoring of acute gastroenteritis diarrhea in children aged 5 and below in the Nantong area of Jiangsu. The detection rate of SaV was 3.93%, which is slightly lower than the positivity rate of SaV infection in children aged 5 and below in the eight provinces and cities in China from 2012 to 2014 (4.38%) [29] and in Hangzhou, China from 2012 to 2019 (4.22%) [30]. The prevalence of SaV in this region shows a seasonal distribution, with the highest intensity occurring during the late autumn and winter seasons. This finding is similar to the discovery made by Spanish scholars [31], but it contradicts the research results from scholars in Tibet and Urumqi in China [32]. This could be because Spain and Nantong, China are located at latitude 36° N and latitude 32° N, respectively. They are both coastal areas with similar climate characteristics. On the other hand, Tibet and Urumqi are located in inland regions of China and belong to the high latitude areas. They have a significant difference in latitude compared to this region, resulting in major differences in climate characteristics.

The GI.1 genotype of SaV is the most widely prevalent genotype worldwide. In this study, the strain sequences obtained showed that the GI.1 genotype accounted for 88.89% (8/9), while the GI.6 genotype accounted for 11.11% (1/9), which is consistent with the prevailing trend. It is worth noting that the GI.6 genotype has been less detected and reported by researchers in previous studies. Apart from Japanese scholars who discovered this genotype as the dominant strain in the local area in 2004–2005 [33], all other reported GI.6 genotypes in endemic areas are considered as low-prevalence strains [34,35,36,37].

We compared the nucleotide consistency of the local strains of both types with their corresponding prototype strains and found that the consistency was about 90%, indicating significant changes. The SaV ORF1 encodes a polyprotein and the major capsid protein VP1, while the polyprotein can be cleaved by the Pro (protease) into non-structural proteins NS1, NS2, NS3 (helicase), NS4, NS5 (viral protein, VPg), Pro, and RdRp (RNA-dependent RNA polymerase). The functions of RdRp, Pro, helicase, and VPg have been identified through sequence similarity with homologous proteins from small RNA viruses and other positive-sense single-stranded RNA viruses [10]. By comparing the amino acid consistency in the protein-coding regions of the local strains and prototype strains, we found that, except for NS1, the nucleotide mutations in the protein-coding regions were primarily synonymous mutations, with fewer non-synonymous mutations. This suggests that, except for NS1, although the protein-coding regions are constantly evolving through nucleotide mutations, they rarely involve changes in amino acids that would affect protein function. Some studies have suggested that mutations in the NS1 coding sequence of caliciviruses may affect tissue tropism, virulence, and epidemiological adaptability [38], which may explain the higher proportion of non-synonymous mutations in the NS1 coding region.

This study included all GI.1 and GI.6 SaV strains for analysis. Evolutionary analysis revealed that the obtained strain sequences in this study were located in the same evolutionary branch in the phylogenetic trees constructed based on the complete genome, VP1, ORF1, and ORF2, indicating no evidence of recombination. The comparison of molecular evolution rates between GI.1 and GI.6 SaV showed that the molecular evolution rate of GI.6 was significantly higher than GI.1. The reason for this difference may be due to the limited number of GI.6 SaV sequences that are currently available for analysis, resulting in a certain degree of deviation in the analysis results from the true situation. In future studies, we will focus more on the detection of rare SaV genotypes, providing data reference for future SaV research.

Based on MCMC inference of evolutionary parameters, GI.1 SaV split into two lineages, Lineages I and II, around 1949. The most recent common ancestor of Lineage I (1959) predates that of Lineage II (1974), suggesting that Lineage I likely diverged earlier than Lineage II. Through comparative analysis of amino acid differences between Lineages I and II, we identified 21 amino acid sites that are the main drivers of the division of GI.1 SaV into two lineages. When compared to the prototype strain HM002617, Lineage II had more amino acid changes at these 21 sites than Lineage I. Furthermore, these changes were predominantly concentrated in the non-structural protein region of Lineage I, while in Lineage II, they were scattered across multiple functional regions of the virus. This indicates that there have been significant changes in the evolutionary direction of genes between these two lineages, further supporting the notion that the molecular evolution rate of Lineage II is slightly higher than that of Lineage I.

Based on the population dynamics analysis of the global GI.1 and GI.6 strains of SaV, it is evident that the GI.6 strain has consistently shown a stable trend, suggesting that it has difficulty becoming the dominant genotype in China and globally. On the other hand, the effective population size of the two lineages of the GI.1 strain shows a competition suppression phenomenon (as illustrated in Figure 6B,D), with the most significant occurrence observed between 2002 and 2013. This indicates that the two lineages of GI.1 SaV strains may alternately become the dominant strains during the epidemic period.

Overall, this study conducted an analysis of the molecular epidemiological characteristics and genetic evolutionary characteristics of prevalent SaV in Nantong, China. The GI.6 strain sequence found in our study filled the gap in the Genbank database of our country and was the first. It has laid a solid foundation for disease prevention, control, and pathogen tracing related to the SaV.

The limitation of this study is that it was not possible to obtain all sample sequences that tested positive for SaV during the monitoring period. This may be attributed to quality issues in specimen collection or the susceptibility of RNA viruses to degradation. In the future, we will make efforts to optimize the process of specimen collection and ensure proper preservation and transportation of collected specimens. We believe that these improvements will lead to an increased success rate in sequencing.

## Figures and Tables

**Figure 1 viruses-16-00146-f001:**
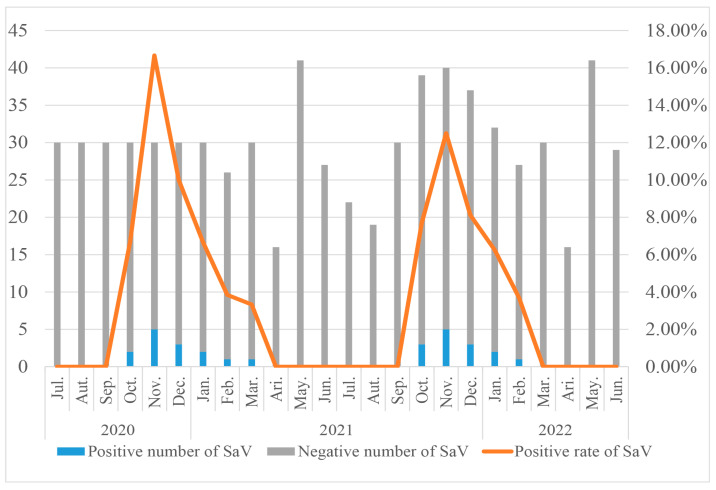
General situation of SaV infections in children under 5 years old with acute gastroenteritis in the Nantong area.

**Figure 2 viruses-16-00146-f002:**
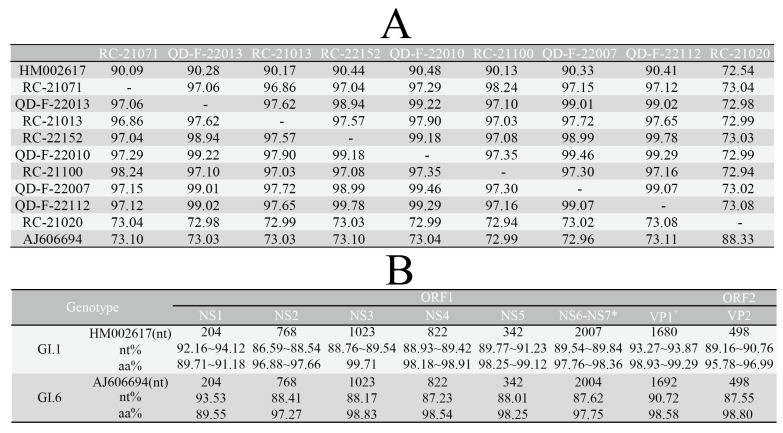
Evolutionary divergence between nine local strains and corresponding genotype prototype strains. (**A**) Nucleotide identity (%) between local strains based on the whole genome and corresponding genotype prototype strains; (**B**) identity analysis between local strains and corresponding genotype prototype strains based on different genes; *, in the NS6-NS7 region, eight local strains of the GI.1 type showed a deficiency in the 590th amino acid position compared to the prototype strain HM002617; ^+^ in the VP1 region, GI.1 SaV has 1680 nucleotides encoding 559 amino acids, while GI.6 SaV has 1692 nucleotides encoding 563 amino acids.

**Figure 3 viruses-16-00146-f003:**
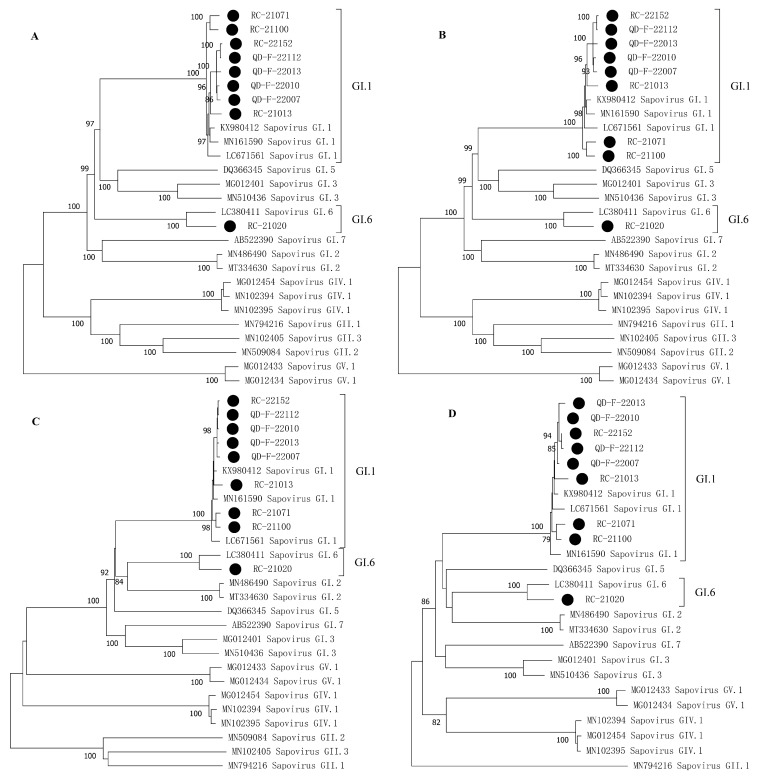
Phylogenetic analysis of SaV whole genome and different genes. The strain sequences obtained in this study are highlighted with ●. (**A**) Phylogenetic tree constructed based on SaV whole genome sequence; (**B**) phylogenetic tree constructed based on SaV NSP (ORF1) gene; (**C**) phylogenetic tree constructed based on SaV VP1 gene; and (**D**) phylogenetic tree constructed based on SaV VP2 (ORF2) gene phylogenetic tree.

**Figure 4 viruses-16-00146-f004:**
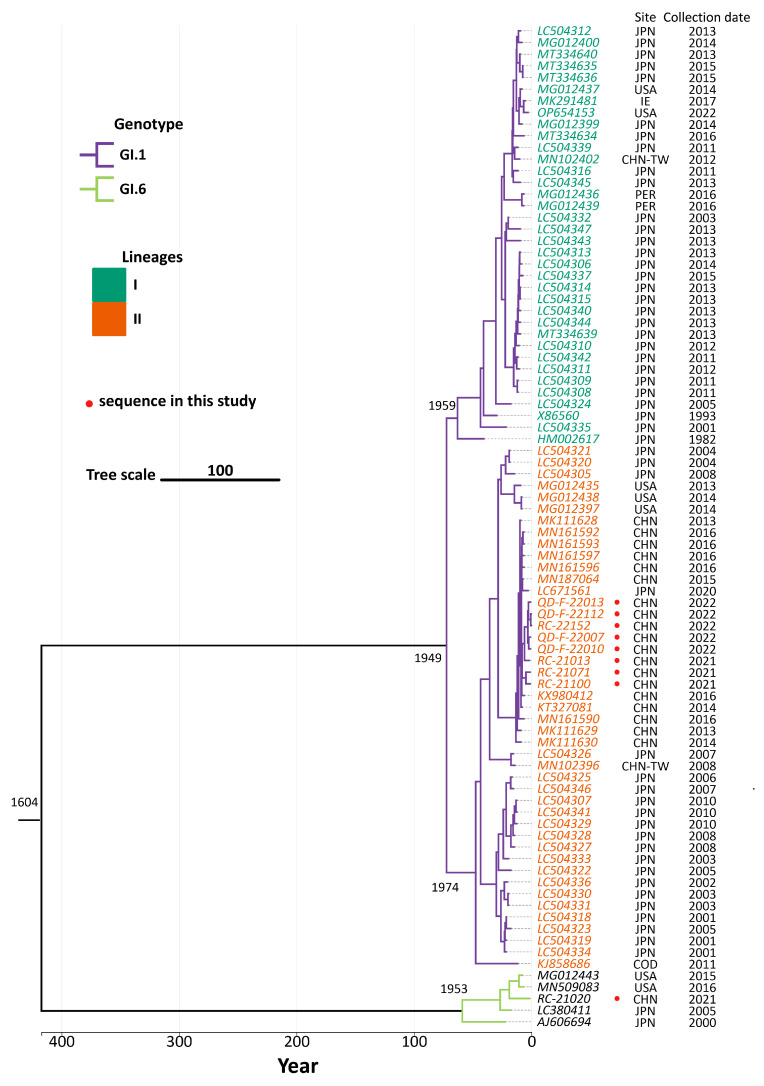
MCC based on the SaV whole genome. Branches with different colors represent different genotypes; leaves with different colors represent different lineages; and the SaV sequence obtained in this study is highlighted in red ●.

**Figure 5 viruses-16-00146-f005:**
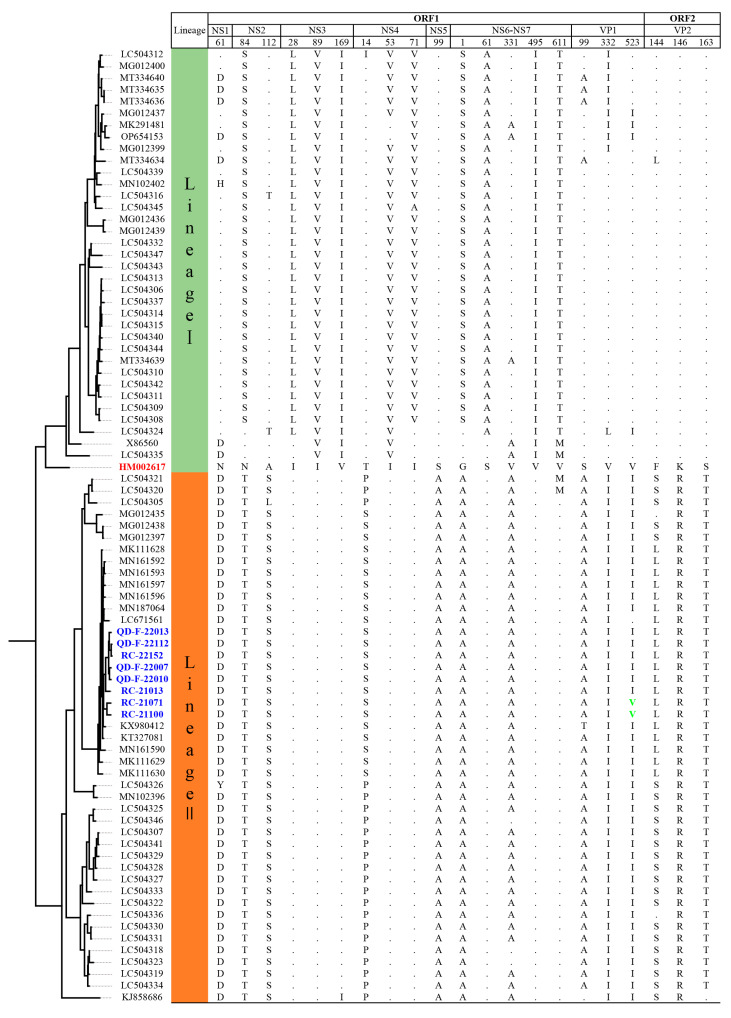
Schematic diagram showing the main differences in amino acids between two lineages of GI.1 SaV. The red numbered markers indicate the prototype strain of GI.1 SaV; the blue numbered markers indicate the GI.1 SaV obtained in this study.

**Figure 6 viruses-16-00146-f006:**
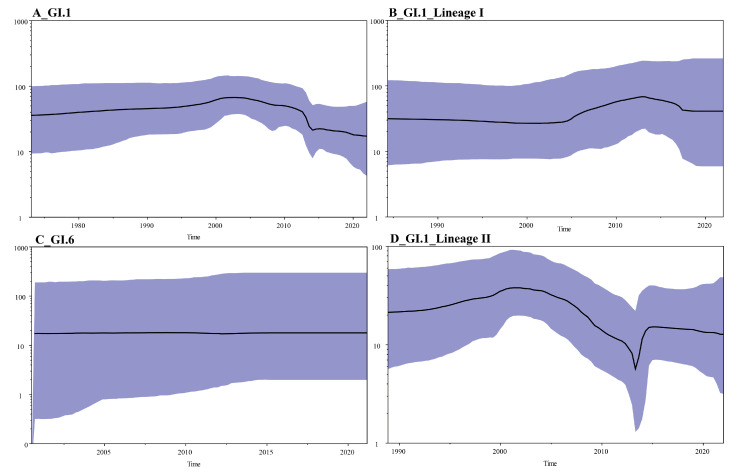
Bayesian Skyline Plot (BSP) of SaV. (**A**), all GI.1 SaV (**B**), Lineage I of GI.1 SaV (**C**), Lineage II of GI.1 SaV and (**D**), all GI.6 SaV. The *x*-axis represents the time in years; the *y*-axis represents the effective population size; the black solid line represents the median posterior value; and the blue area represents the 95% highest probability density (HPD) interval.

**Table 1 viruses-16-00146-t001:** The optimal nucleotide substitution models for each clade.

Region	Clade	Nucleotide Substitution Models
Whole genome	GI.1-Lineage I	GTR + F + I + G4
	GI.1-Lineage II	GTR + F + I + G4
	GI.1	GTR + F + G4
	GI.6	GTR + F + G4
	Total	GTR + F + I + G4

**Table 2 viruses-16-00146-t002:** Evolution parameters inferred based on MCMC.

Clade	Local Strain	Reference Strain	Nucleotide Substitution Rate [Substitutions/Site/Year (95%HPD)]	Date of MRCA
Lineage I	0	36	1.603 × 10^−3^ (6.597 × 10^−4^~2.737 × 10^−3^)	1959
Lineage II	8	37	2.082 × 10^−3^ (1.569 × 10^−3^~2.653 × 10^−3^)	1974
GI.1	8	73	2.092 × 10^−3^ (1.484 × 10^−3^~2.739 × 10^−3^)	1949
GI.6	1	4	6.881 × 10^−3^ (1.874 × 10^−5^~2.930 × 10^−2^)	1953
total	9	86	2.099 × 10^−3^ (1.143 × 10^−3^~2.761 × 10^−3^)	1604

## Data Availability

The data presented in this study are available on request from the corresponding authors.

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
