# Peer review of "Genomic Characterization and Molecular Evolution of Sapovirus in Children under 5 Years of Age"

_viruses, 2024, doi:10.3390/v16010146_

Round 1

Reviewer 1 Report

Comments and Suggestions for Authors

This paper describes Sapoviruses - members of virus family Caliciviridae. They are not a common cause of acute gastroenteritis so they are not well studied. There are limited papers describing Sapoviruses, so this one is important and interesting for researches that work in the same area. But there is a lot of information that should be provided when you write a paper describing genomes of viruses. First of all you must provide in chapter results parameters of your raw data from NGS platform, such as number of reads their length distribution and quality, then you should describe contigs that you got after analysis (length and coverage). You should correctly describe how you trim your data and what method you used to get contigs (mapping on a reference or de novo assembly). Chapter Introduction should be improved by adding information about virus proteins and their functions encoded in genome, because you describe evolution information about them. Also, you have mentioned in Abstract that no evidence of recombination have been found, but you made this conclusion only by watching on phylogenetic trees with limited number of isolates. To declare that you have not observed recombination events you should use special software (GARD or RDP for example). And finally materials and methods should be written in past tenses not in present. Pay attention that from line 67 to line 129 you should rewrite your text in past form. Also it will be more informative if you will add information about number of nucleotides in every coding region on a figure 2. Also I would say that it is hard to make a conclusion for GI.6 subtype using only four isolates, because it is not relevant. From line 238 to 245 you appeal to figure 5 (A-D), but figure 5 don’t have division by letters (probably you appeal to figure 6). Moreover in caption to figure 6 you should describe what are letters A-D. And the last, from my point of view another type of histogram should be used in figure 1, you may summarize in one bloc positive+negative=number of tested. Histograms will be more compact. After all changes this paper can be published.

Comments on the Quality of English Language

The quality of English is ok, except the wrong use of tenses in materials and methods.

Reviewer 2 Report

Comments and Suggestions for Authors

The manuscript that I reviewed “Genomic characterization and molecular evolution of Sapovirus in children under 5 years of age” is a study aimed to perform a molecular investigation on sapoviruses by screening 712 stool samples collected from children under the age of 5 with acute gastroenteritis. A total of 28 samples were identified as positive. By high-throughput sequencing of samples with ct < 30, nine complete genomes were generated, eight G1.1 and one G.1.6. The Authors performed an interesting analysis of sequence identity, phylogeny, and genetic evolution.

General Comments:

Overall, the study conducted is interesting and well written. The statements described in the manuscript are supported by presented data. The study represent an important molecular investigation, above all considering the generation of nine complete SaV genomes, contributing to the study of the genetic characteristics and evolution of these viruses.

However, I have only some points that need to be discussed.

1) Line 41, I suggest to the Authors to replace “…in the family of caliciviruses.” with “….in the Caliciviridae family”.

2)Line 61, “ACE” is “AGE”?

3)Line 58-64, In the “Materials and Methods” section, “Source of the specimen” subsection, I suggest to the Authors to describe in detail, maybe with a table, the sampling to clearly understand the prevalence results and figures. (e.g., how many samples for each mounth….)

4)Line 275-280, I suggest to include this description in the “Introduction” section.

Reviewer 3 Report

Comments and Suggestions for Authors

This is a report on the genomic characterization and molecular evolution analysis of a sapovirus isolated in china.

Overall, the manuscript is well written.

I have several suggestions which may improve the quality of the manuscript.

1. If possible, please describe the cut-off values for nt and aa identity between genotypes and genogroups.

2. Figure 1B seems unnecessary.

3. Please write the genotype of the prototype in Figure 1A.

4. Please write in Figure 5 instead of Figure 4. (page 4, 224 lane).

5. What is limitation of the study? No limitations of the study are indicated.

6. Please describe more clearly the significance of the results.
